# Survival analysis of under-five mortality and associated risk factors using survival analysis approaches

**Abdul-Karim Iddrisu** [1]*, **Emmanuel Boanyo** [2]

**1** Department of Statistics, School of Social Sciences, Faculty of Social Sciences, University of Botswana, Gaborone, Botswana, **2** Department of Mathematics and Statistics, School of Science, University of Energy and Natural Resources, Sunyani, Ghana

☉ These authors contributed equally to this work.
* karim@aims.ac.za

## Abstract

The risk of under-five mortality is a vital measure of healthcare system performance and directly reflects progress toward Sustainable Development Goal (SDG) 3.2, which targets the elimination of preventable deaths among newborns and children under-five, aiming to reduce mortality rates to at least 25 per 1,000 live births by 2030. While Ghana has made notable progress in lowering child mortality in recent decades, the current rates remain above this benchmark. Identifying the predictors of under-five mortality is therefore critical for shaping evidence-based policies and targeted interventions that can accelerate progress toward SDG 3 and improve child health outcomes. To explore these predictors, we employed advanced survival modeling techniques. The conventional Cox-proportional hazards (Cox-PH) model assumes constant covariate effects over time, but violations of this assumption can lead to biased results. To address this, we used the extended Cox-PH model, which accommodates time-varying effects. Data were drawn from the 2022 Ghana Demographic and Health Survey (GDHS), based on a stratified two-stage cluster sampling design. Since under-five deaths are relatively rare (<10%), traditional models may yield unstable hazard ratios. We therefore applied Bayesian survival analysis to obtain more stable estimates and incorporated multilevel survival modeling to account for unobserved heterogeneity within the DHS sampling structure. Results showed that male children (HR = 1.20, 95% CI: 1.11–1.30) and twins (HR = 2.90, 95% CI: 2.51–3.34) faced higher mortality risk. Caesarean delivery (HR = 1.60, 95% CI: 1.08–2.37) and larger birth size also increased hazards. In contrast, term birth (HR = 0.16, 95% CI: 0.14–0.19), maternal education, and higher household wealth were protective. Children requiring special attention after delivery had improved survival (HR = 0.57, 95% CI: 0.38–0.89). Strengthening maternal and newborn care, coupled with addressing socioeconomic inequalities, is essential to reducing child mortality and achieving Ghana's SDG 3.2 targets.

**Data availability statement:** The data used in this study are publicly available at https://dhsprogram.com/methodology/survey/survey-display-598.cfm.

**Funding:** The author(s) received no specific funding for this work.

**Competing interests:** The authors have declared that no competing interests exist.

## Introduction

Infant mortality remains a major global public health concern and a key indicator of healthcare quality and socioeconomic development [1]. Despite medical advancements, significant disparities persist, particularly between high- and low-income countries [2]. In 2020, over 5 million children under five died, including 2.4 million newborns, with most deaths being preventable [1]. The neonatal period, accounting for nearly half of under-five deaths, remains the most critical for survival. Although neonatal and infant mortality rates have declined significantly since 1990, 2.3 million newborns and 13,400 children under five still died daily in 2022, underscoring the urgency of targeted interventions to reduce preventable child deaths [1]. Persistent healthcare inequities highlight the need for strengthened maternal and child health services, especially in resource-limited settings.

Sub-Saharan Africa (SSA) experiences the highest burden of infant mortality, accounting for over half of global infant deaths [1]. The region's infant mortality rate stands at 47 deaths per 1,000 live births, significantly higher than the 5 per 1,000 observed in high-income countries [2]. Among the ten countries with the highest infant mortality rates, nine are in Africa, with Somalia leading at 83.6 deaths per 1,000 live births, followed by the Central African Republic, Equatorial Guinea, and Sierra Leone [3]. This disparity is driven by multiple factors, including economic development, healthcare infrastructure, governance, and education play critical roles. Key medical causes include preterm birth complications, birth asphyxia, pneumonia, diarrheal diseases, malaria, and HIV/AIDS [4,5]. Sociodemographic factors such as maternal age, multiple births, low birth weight, place of delivery, breastfeeding practices, and birth spacing further contribute to infant mortality risk [6]. Addressing these challenges requires comprehensive public health strategies, improved healthcare access, and targeted interventions to reduce preventable infant deaths in the region.

Although Ghana has made progress in reducing infant mortality, the country faces significant challenges in achieving the Sustainable Development Goal (SDG) 3.2 target of reducing neonatal mortality to 12 per 1,000 live births and under-five mortality to 25 per 1,000 live births by 2030 [7]. The current infant mortality rate in Ghana is approximately 32 deaths per 1,000 live births, which, while an improvement, remains a concern [7]. Key factors contributing to this issue include inadequate antenatal care, limited postnatal monitoring, insufficient breastfeeding support, human resource challenges, and access to skilled birth attendants [8]. Other challenges include poor maternal nutrition, infectious diseases, regional healthcare disparities, inadequate infrastructure, limited access to essential medicines, and cultural beliefs [9]. Rural areas experience particularly high infant mortality rates due to limited healthcare infrastructure and poor transportation, which delay access to critical maternal and neonatal care [10,11].

Accurate estimation of child mortality risk is crucial for developing effective policies and intervention strategies. Several statistical methods have been used to study child mortality and associated risk factors using negative binomial [12], Bayesian spatio-temporal framework [13–15], and geostatistical survival models [16,17].

In some study designs, following study participants for a specific duration is necessary to observe how the outcome of interest changes or develops. Such study designs are common in Biostatistics and medical research, known as a longitudinal study design [1–5]. In the context of longitudinal study design, which is frequently employed in repeated measures designs, data points are repeatedly collected for a response at some selected schedule visits. Apart from the longitudinal measurements, it is common in the majority of medical research scenarios to document information related to time-to-first occurrence of an event of interest, such as the time until death, recovery from an illness, progression to a disease, and so on. Survival data is a common term used to describe the time-to-event information [6,7].

Longitudinal and survival data can be analyzed independently, and well-established methods for such separate analyses can be found in the literature. For instance, the mixed-effects models [8] are employed for modelling the longitudinal aspect of the data [1–5], while Cox-proportional hazard models [9,10] are utilized for the analysis of survival outcomes. However, in a situation where the progression or patterns observed in the recurring measurements of the response variable within the same subject are likely to be impacted by the events occurrence, rendering them endogenous [11], a potential solution to this issue is to us a joint model for longitudinal and survival [12,18], as it mitigates bias in parameter estimation and enhances the efficiency of parameter estimates.

The Cox-PH model is a widely utilized statistical tool in survival analysis, particularly in assessing the impact of various risk factors on time-to-event outcomes, such as under-five mortality [13–17,19,20]. This model estimates hazard ratios (HRs), indicating the effect of predictor variables on the hazard or risk of an event occurring at a particular time point [15]. Globally, the Cox proportional hazards model has been instrumental in identifying determinants of under-five mortality. For instance, a study employing this model highlighted that half of global child mortality in 2018 occurred in five countries, underscoring significant disparities in child survival rates [17]. In SSA, the model has been extensively applied to assess the impact of socio-economic and demographic factors on under-five mortality [20]. Similarly, research in Ethiopia utilized the Cox model to reveal that children residing in Addis Ababa had a lower hazard of death compared to other regions, emphasizing regional disparities within the country [14]. Research in Uganda employed random survival forests alongside Cox models to understand determinants of under-five mortality [19].

The Cox-PH model has been widely used to study under-five mortality in Ghana [16]. However, based on the available literature, there is limited evidence of the application of extended versions of this model; such as those incorporating time-dependent covariates or frailty effects in this context. For instance, a study analyzing data from the 2017 Ghana Maternal Health Survey employed the standard Cox-PH model to examine geographic disparities in under-five mortality [16]. While standard Cox-PH models have been extensively utilized, the application of extended Cox models such as those incorporating time-varying covariates or frailty components, appears to be less common in studies focusing on under-five mortality in Ghana. Research employing these advanced modelling techniques could provide deeper insights into the temporal dynamics and unobserved heterogeneity associated with under-five mortality in the Ghanaian context. This study seeks to investigate the hazard of time-to-death among under-five children using the multivariable extended Cox-PH model [10,21,22], the Bayesian survival analysis [12–14,16,19,20] and we accounted for potential unobserved heterogeneity using a multilevel survival modelling approach [23–27]. In this study, we aimed to estimate and compare survival probabilities between male versus female while adjusting for the impact of some selected potential predictors of survival probabilities among under-five children. The key hypotheses that derive this study are (1) the hazard of the event significantly differ between males and females and (2) some variables have significant effects on hazard under-five mortality.

## Materials and methods

### Design

The data utilized in this study were sourced from the 2022 Ghana Demographic and Health Survey (GDHS 2022), publicly available at https://dhsprogram.com/methodology/survey/survey-display-598.cfm. A total of 34,663 under-five children,

collected as part of a nationally stratified, representative sample was used in this study. A total of 18,450 households were selected across 618 clusters, resulting in 15,014 interviewed women (aged 15–49 years) and 7,044 interviewed men (aged 15–59 years), with male respondents drawn from every second household (Fig 1). The sampling procedure employed in the GDHS 2022 followed a stratified two-stage cluster sampling design, ensuring national representation across urban and rural areas and each of the 16 regions for key DHS indicators. First stage: A total of 618 clusters were selected using probability proportional to size (PPS) within urban and rural areas of each region. The final selection of clusters was conducted through systematic random sampling to ensure equal probability within designated strata. Second stage: A comprehensive household listing and map updating operation were carried out within the selected clusters to generate a complete sampling frame. A systematic random selection was then performed to identify the final household sample. This methodology was designed to ensure the representativeness and reliability of findings at both national and regional levels, facilitating robust statistical analysis of under-five mortality and associated factors.

## Study variables

**Survival variable.** The survival variable is **death** (takes the values of 1 if child is a dies and 0 if child is alive or censored). The survival time variable is time from birth until the event of interest (death of the child) occurs or until the end period of the study (59 months).

## Predictors

Guided by existing literature [12–14,16,18,19], a range of potential predictors of child mortality were identified and considered. The potential predictors considered in this study are **sex** (takes values of 1 if child is a male and 0 if child is a female). The predictors are **ever-had a terminated pregnancy** (takes values of 1 if child's mother has ever terminated pregnancy and 0 if mother has never terminated), **twins** (takes value of 1 if child is a twin and 0 if single), **gestation period** (takes values of 1 if gestation period is at least 9 months and 0 if gestation period is less than 9 months), **tetanus vaccination before birth** (takes values of 1 if mother receives tetanus vaccine once before birth, 2 if mother receives tetanus vaccine 2–7 times before birth and 0 if mother never receives tetanus vaccine before birth), **parental** care (takes values of 1 if child's mother received parental care and 0 if mother never received parental care), **wanted pregnancy when became pregnant** (takes values of 1 if child's mother wanted pregnancy later when she became pregnant, 2 if child's mother did not want pregnancy anymore when she became pregnant and 0 wanted pregnancy then when she became pregnant), antenatal care (**ANC**) (takes values of 1 if mother attended ANC at least 4 times and 0 if mother

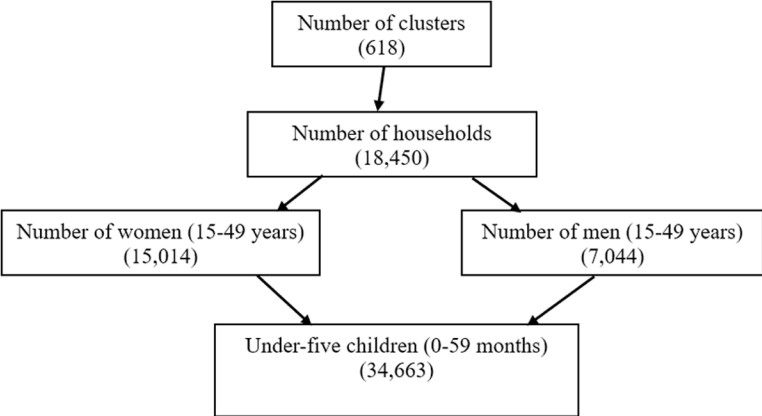

**Fig 1. Flow-chart representing data collection procedure.**

attended ANC less than 4 times), **caesarean section (CS)** (takes values of 1 if child is delivered through **caesarean** section and 0 if child was delivered through normal delivery), **child size** (takes values of 1 if child size is categorized as large, 2 if child size is categorized as average and 0 if child size is categorized as small), **malaria in pregnancy** (takes values of 1 if child's mother was diagnosed with malaria during pregnancy and 0 if child's mother was never diagnosed with malaria during pregnancy), **mother needs intestinal drug during pregnancy** (takes values of 1 if child's mother needed intestinal drug during pregnancy and 0 if child's mother never needed intestinal drug during pregnancy), **child needs medical attention after delivery** (takes values of 1 if child needed medical attention after delivery and 0 if child did not need medical attention after delivery), **geographical location** (takes values of 1 if child lives in an urban area and 0 if child lives in rural area), **maternal educational level** (takes values of 1 if child's mother has obtained higher education, 2 if mother has obtained secondary education, 3 if mother has obtained primary education, and 0 if mother has no education), **wealth** index (takes values of 1 if wealth index is described as richer or richest, 2 if middle and 0 if poorer or poorest), **mother smoked during pregnancy** (takes values of 1 if child's mother smoke cigarette and 0 if mother never smoke cigarette). Results on only significant predictors would be presented in discussed. Statistical significance is defined as p-value <0.05 [20].

## Statistical methods

Kaplan-Meier curves were utilized to graphically compare the survival of under-five children across different groups, with the support of the Log-rank test to assess the statistical significance of survival differences between groups. Semi-parametric [15,17,21,28–31] and extended [22,32–35] Cox-PH model with multiple variables were subsequently applied. We then applied the Bayesian survival model, multilevel survival model to the data, compared the results from the extended Cox-PH model, Bayesian survival model and the multilevel survival model, and results reported.

## Semi-parametric Cox-PH model

The Cox-PH model is a widely used method for estimating the effects of covariates on the hazard of an event [36,37]. The model is specified as: $h(t|X) = h_0(t)\exp(\beta_1 X_1 + \beta_2 X_2 + \ldots + \beta_k X_k)$, where $h(t|X)$ represents the hazard function at time $t$ given $k$ explanatory variables $X_1, X_2, \ldots, X_k$, and $\beta_1, \beta_2, \ldots, \beta_k$ are the corresponding regression coefficients quantifying the effects of these variables on the hazard. The term $h_0(t)$ denotes the baseline hazard function, representing the hazard when all covariates are zero. Unlike parametric survival models, the Cox-PH model does not assume a specific functional form for the baseline hazard $h_0(t)$, making it a semi-parametric approach. The key assumption of the Cox-PH model is the proportional hazards assumption, which states that the hazard ratios between different levels of the independent variables remain constant over time.

## Assessing the assumption of the Semi-parametric Cox-PH model

The proportional hazards assumption states that the hazard functions for different groups must remain proportional over time, meaning that the effect of covariates on the hazard function remains constant throughout the observation period. This implies that the hazard ratio remains stable over time. Verifying this assumption is essential before applying Cox-PH regression analysis. Several methods are available to assess the validity of this assumption [38]. These include: (1) inspecting Kaplan–Meier survival curves, where crossing curves or instances where one curve declines while another plateaus suggest potential violations, and (2) analyzing scaled Schoenfeld residuals, which provide both statistical tests and graphical diagnostics for assessing the proportional hazards assumption [35]. In this study, we examined scaled Schoenfeld residuals to evaluate the validity of the proportional hazard assumption. When a covariate was found to violate the assumption of a constant effect over time, an extended or time-dependent Cox-PH model was applied to account for time-varying effects [22,32,33].

> **Schoenfeld residuals**: A type of statistical check used to see if the effect of a variable on the outcome changes over time. If the effect stays the same, the model fits well; if not, it may suggest the assumption of constant risk over time is violated.

> **Time-varying effects**: When the effect of a risk factor is not constant but changes during the follow-up period. For example, a risk factor might have a strong effect shortly after the follow-up begins, but a weaker effect later.

### Extended Cox-proportional hazard model

When covariates in the Cox-PH model violate the assumption of constant effects over time, an extended or time-dependent Cox model is employed [22,32–34]. To determine whether a variable $X$ exhibits a time-varying impact on the event hazard, a time-dependent term is constructed by interacting the predictor $X$ (which may be continuous or categorical) with a function of time $f(t)$ (e.g., $t, t^2, \log(t), \sqrt{t}$, etc.). By incorporating this interaction term into the model, the hazard function is reformulated as:

$$h_X(t) = h_0(t)\exp\left(\beta X + \gamma X f(t)\right) = h_0(t)\exp\left(\sum_{p=1}^{P} \beta_p X_p + \sum_{p=1}^{P} \gamma_p X_p f(t)\right),$$

> **Interaction term**: Statistical term that test whether the effect of one factor depends on another. For instance, whether the effect of vaccination on survival is different for male versus female.

where $\gamma$ quantifies the change in the effect of $X$ as $f(t)$ progresses. Consequently, the hazard ratio, defined as $HR(t) = h_{(X+1)}(t)/h_X(t)$, reflects the timedependent change in hazard rates associated with a one-unit increase in $X$. When $\gamma > 0$ $\gamma < 0$, the hazard ratio decreases.

 Covariates identified as violating the proportional hazards assumption must have their interactions with time included in the Cox-PH model. An alternative method for handling time-varying coefficients is to use a step function, for instance, $f(t) = I(t \geq t_0)$, where $t_0$ represents a threshold time. This approach partitions the analysis period into distinct intervals, with the Cox model stratified by these time segments, thereby allowing the effect of baseline covariates to vary over time. In this study, we address the violation of the proportional hazards assumption by employing the step function approach, stratifying the observation period into groups of time intervals.

### Bayesian survival analysis

Under-five deaths are rare in DHS samples, typically representing less than 10% of observations, which can lead to biased estimates, unstable hazard ratios, or inflated Type I errors when using standard survival models [31,39,40]. Rare-event outcomes can also make hazard ratio estimates sensitive to outliers and potentially mislead statistical inference [41]. To address this issue, we employed a Bayesian survival analysis, which stabilizes hazard ratio estimates and provides more reliable inference in the context of low event rates [40,41]. Bayesian models have been shown to effectively mitigate the biases and inflated estimates that can arise in classical Cox proportional hazards models when events are sparse [40–43].

### Multilevel survival model: frailty models

Given the hierarchical nature of DHS data, with individuals nested within households and households nested within clusters or regions, we accounted for potential unobserved heterogeneity using a multilevel survival modelling approach

[23–27]. Specifically, we included random effects (frailty terms) for children nested within clusters, clusters within regions at the cluster and regional levels to capture unobserved variation in risk across clusters and regions.

## Results

We now consider analysis of the survival data. In this study, none of the variables from the Ghana Demographic and Health Survey (GDHS) dataset used in the analysis contained missing values (see GDHS dataset at https://dhsprogram.com/methodology/survey/survey-display-598.cfm, or available at Zenodo https://doi.org/10.5281/zenodo.16538777). Both the outcome (child survival status) and all covariates were complete for all observations. We encourage that, in situations where data are incomplete [31,40–42], as is often the case with DHS datasets, missing values should be addressed using appropriate methods such as multiple imputation under the Missing at Random (MAR) assumption, while sensitivity analyses under Not Missing at Random (NMAR) assumptions (e.g., pattern-mixture [41,43], selection [41,42], or shared-parameter [23,43] models) are recommended as best practice [23,24].

### Kaplan-Meier curves and log-rank test

We estimated and compared the survival probability of under-five mortality between male and female groups using the Kaplan-Meier method, followed by an assessment of the significance of survival differences between the groups using the log-rank test. Similarly, we applied these methods to examine the survival probability of under-five mortality across other potential predictors. For the purpose of illustration, Fig 2 displays the Kaplan-Meier curves and log-rank test results for the potential predictors **sex**, **twins**, **wealth** status, and **gestation** period. Similar results for the remaining potential predictors were also generated with the results reported.

Fig 2 illustrates statistically significant differences in the proportion of under-five mortality across various groups. Specifically, significant differences were observed between males and females ($\chi^2_{1,0.025} = 15.07$, p-value<0.001), twins and singletons ($\chi^2_{1,0.025} = 303.73$, p-value<0.001), wealth status groups ($\chi^2_{1,0.025} = 26.65$, p-value<0.001), and between children with a gestation period of at least nine(9) months versus less than nine(9) months ($\chi^2_{1,0.025} = 26.65$, p-value<0.001). Similar results for the remaining potential predictors indicate no significant difference in the survival probability of under-five mortality for children whose mothers who had ever terminated a pregnancy versus whose mothers have never, whose mothers smoked during pregnancy versus whose mothers do not smoke during pregnancy and who live in rural area versus urban area. However, a significant difference in survival probability was observed for the other potential predictors of under-five mortality.

### Results from the semi-parametric Cox-PH model

The Cox-PH model was fitted in *R* version 4.3.1, for the survival outcomes death, using coxph function from survival package [41,42]. We have fitted univariable model and then used significant variables in the univariable model to fit ta multi-variable semi-parametric Cox-PH models to estimate the unadjusted hazard ratios (unaHR) and adjusted hazard ratios (aHR), as presented in Table 1. The unaHR results aligned with the results from the Kaplan-Meier and log-rank methods, indicating that the only none significant predictors of under-five mortality are termination of pregnancy, geographical location, and maternal smoking. Before presenting the final results from the semi-parametric Cox-PH models, we assess whether the assumption of constant covariate effects over time is satisfied.

### Checking violation the Cox-PH model assumption

In this study, we evaluated the proportional hazards (PH) assumption of the Cox model using Schoenfeld residuals, implemented through the **cox.zph** function from the survival package [42,41]. The assessment was conducted for both univariable and multivariable Cox-PH models to examine their validity in modelling under-five mortality. This test evaluates

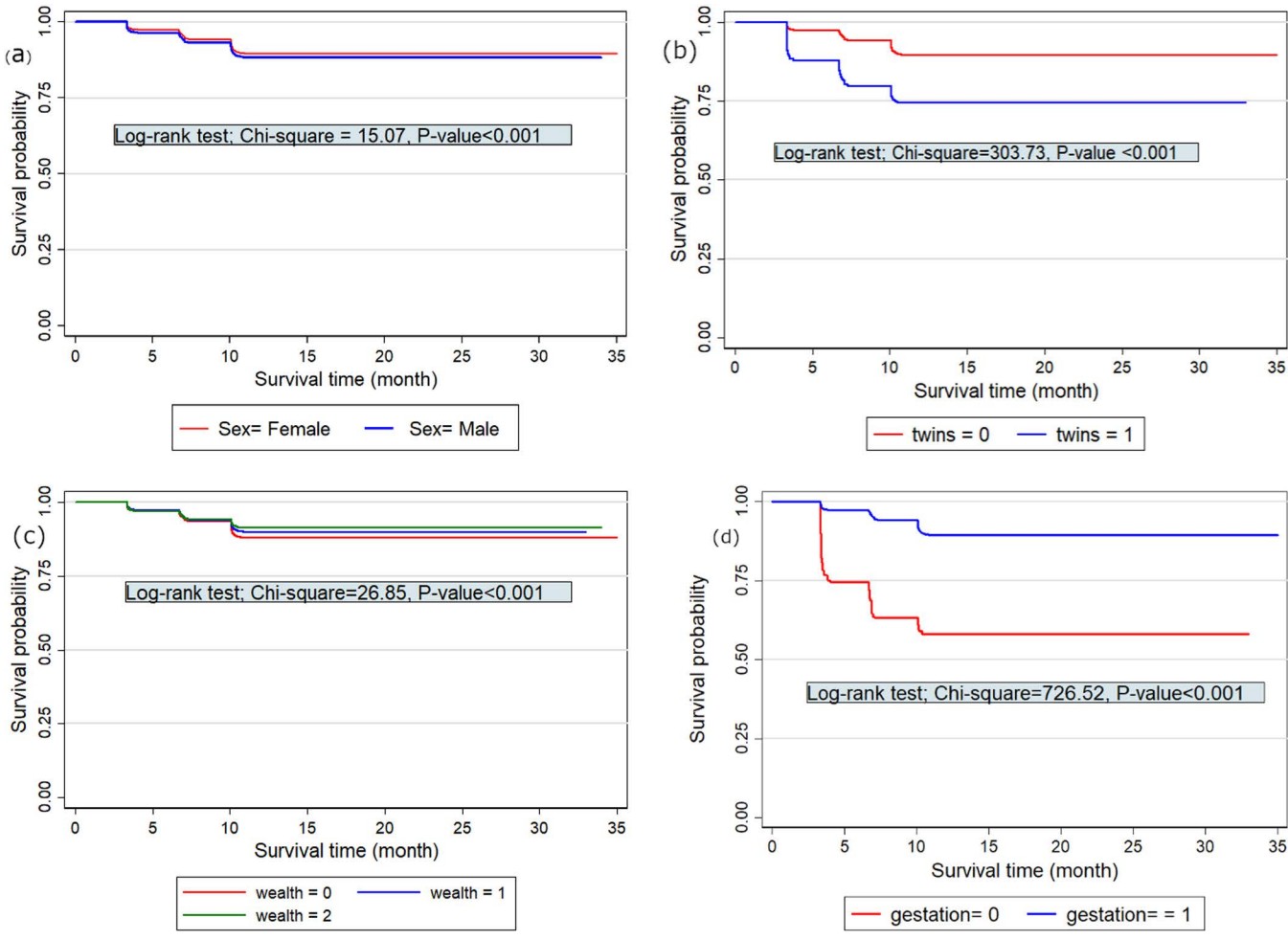

**Fig 2. Kaplan-Meier curves and log-rank estimates for under-five mortality by sex (top-left panel), twins (top-right panel), wealth status (bottom-left panel), and gestation period (bottom-right panel).**

whether the PH assumption holds both globally and for individual covariates by providing p-values, where a p-value < 0.05 indicates a violation of the assumption. As shown in Table 2, in the univariable Cox-PH model, the variables sex, twins, gestational period, child size, malaria during pregnancy, child's need for antenatal care, area, maternal education, and wealth violated the PH assumption. In the multivariable Cox-PH model, the variables sex, twins, gestational period, maternal education, and maternal wealth exhibited violations, as their respective p-values were below 0.05. To ensure valid statistical inference, we account for these violations by fitting an extended Cox-PH model.

### Results from the extended Cox-PH model

We used graphical displays of residuals for covariates that violated the proportional hazards assumption to identify the survival time points requiring stratification. For illustration, Fig 3 presents examples of these graphs for sex, twins, gestation period, and child size in the univariable Cox-PH model. Similar graphs can be generated for covariates violating the assumption in the multivariable Cox-PH model. After determining the appropriate time points, we stratified the survival time for these variables and refitted the Cox-PH model. For instance, in the univariable Cox-PH model, the *sex* variable

**Table 1. Unadjusted and adjusted hazard ratio from the Cox-PH model for time-to-death with standard error (s.e) and 95% confidence interval (95%CI).**

| Time to Death | unaHR. | | | | aHR | | | |
|---|---|---|---|---|---|---|---|---|
| | HR | s.e | 95%CI | P-value | HR* | s.e | 95%CI | P-value |
| **Sex** | | | | | | | | |
| Female (Ref) | – | – | – | – | – | – | – | – |
| Male | 1.18 | 0.05 | (1.08  1.28) | < 0.001 | 1.20 | 0.05 | (1.10, 1.30) | < 0.001 |
| **Ever had a terminated pregnancy** | | | | | | | | |
| No (Ref) | – | – | – | – | – | – | – | – |
| Yes | 0.95 | 0.04 | (0.84, 1.04) | 0.312 | – | – | – | – |
| **Twins** | | | | | | | | |
| Singletons (Ref) | – | – | – | – | – | – | – | – |
| Twins | 3.19 | 0.23 | (2.78, 3.66) | <0.001 | 2.68 | 0.20 | (2.31, 3.08) | <0.001 |
| **Gestation period** | | | | | | | | |
| <9 months (Ref) | – | – | – | – | – | – | – | – |
| $\geq 9$ months | 0.15 | 0.01 | (0.13, 0.18) | <0.001 | 0.19 | 0.02 | (0.17, 0.23) | <0.001 |
| **Tetanus vaccination before birth** | | | | | | | | |
| No vaccination (Ref) | – | – | – | – | – | – | – | – |
| Once | 40.28 | 6.22 | (29.76, 54.51) | <0.001 | 2.55 | 0.66 | (1.54, 4.18) | <0.001 |
| 2-7 times | 33.19 | 5.78 | (23.59, 46.69) | <0.001 | 0.99 | 0.30 | 0.53, 1.79 | 0.918 |
| **Parental care** | | | | | | | | |
| No (Ref) | – | – | – | – | – | – | – | – |
| Yes | 30.33 | 7.89 | (18.22, 50.50) | <0.001 | 0.86 | 0.27 | (0.46, 1.60) | 0.633 |
| **Wanted pregnancy when became pregnant** | | | | | | | | |
| Yes (Ref) | – | – | – | – | – | – | – | – |
| Later | 33.16 | 5.40 | (24.09, 45.63) | <0.001 | 2.04 | 0.40 | (1.39,3.00) | <0.001 |
| No more | 38.69 | 14.68 | (18.39, 81.39) | <0.001 | 0.53 | 0.23 | (0.23,1.23) | 0.140 |
| **Antennal care** | | | | | | | | |
| No (Ref) | – | – | – | – | – | – | – | – |
| Yes | 0.03 | 0.01 | (0.01, 0.07) | <0.001 | 0.68 | 0.30 | (0.29,1.62) | 0.382 |
| **Caesarean section** | | | | | | | | |
| No (Ref) | – | – | – | – | – | – | – | – |
| Yes | 35.90 | 6.26 | (25.52, 50.52) | <0.001 | 1.63 | 0.35 | (1.07,2.49) | 0.024 |
| **Child size** | | | | | | | | |
| Small (Ref) | – | – | – | – | – | – | – | – |
| Large | 33.17 | 4.50 | (25.42, 43.28) | <0.001 | 9.97 | 2.23 | (6.43,15.46) | <0.001 |
| Average | 37.42 | 5.00 | (28.84, 48.57) | <0.001 | 12.03 | 2.58 | (7.90,18.32) | <0.001 |
| **Malaria in pregnancy** | | | | | | | | |
| No (Ref) | – | – | – | – | – | – | – | – |
| Yes | 35.25 | 3.91 | (28.35, 43.82) | <0.001 | 1.34 | 0.37 | (0.73, 2.27) | 0.375 |
| **Mother needs intestinal drug during pregnancy** | | | | | | | | |
| No (Ref) | – | – | – | – | – | – | – | – |
| Yes | 35.70 | 5.27 | (26.72, 47.68) | <0.001 | 1.34 | 0.32 | (0.84,2.14) | 0.221 |
| **Child needs medical attention after delivery** | | | | | | | | |
| No (Ref) | – | – | – | – | – | – | – | – |
| Yes | 29.03 | 4.01 | (22.15, 38.05) | <0.001 | 1.02 | 0.23 | (0.65,1.60) | 0.924 |

*(Continued)*

**Table 1.** (Continued)

| Time to Death | unaHR. | | | | aHR | | | |
|---|---|---|---|---|---|---|---|---|
| | HR | s.e | 95%CI | P-value | HR* | s.e | 95%CI | P-value |
| **Geographical location** | | | | | | | | |
| Rural (Ref) | – | – | – | – | – | – | – | – |
| Urban | 0.99 | 0.04 | (0.91, 1.08) | 0.880 | – | – | – | – |
| **Maternal educational level** | | | | | | | | |
| No education (Ref) | – | – | – | – | – | – | – | – |
| Higher | 0.56 | 0.09 | (0.41, 0.76) | <0.001 | 0.51 | 0.09 | (0.37, 0.71) | <0.001 |
| Secondary | 0.78 | 0.04 | (0.71, 0.86) | <0.001 | 0.76 | 0.04 | (0.68,0.84) | <0.001 |
| Primary | 1.01 | 0.06 | (0.91, 1.13) | 0.861 | 1.01 | 0.06 | (0.90,1.13) | 0.845 |
| **Wealth index** | | | | | | | | |
| Poor (Ref) | – | – | – | – | – | – | – | – |
| Rich | 0.85 | 0.05 | (0.76, 0.95) | 0.004 | 0.87 | 0.05 | (0.77,0.98) | 0.024 |
| Middle | 0.76 | 0.04 | (0.68, 0.86) | <0.001 | 0.81 | 0.06 | (0.71,0.93) | 0.003 |
| **Mother smoked during pregnancy** | | | | | | | | |
| No (Ref) | – | – | – | – | – | – | – | – |
| Yes | 1.30 | 0.27 | (0.86, 1.96) | 0.251 | – | – | – | – |

**Table 2.** Checking proportional hazards assumption.

| Variable | Unadjusted Cox-PH | | Adjusted Cox-PH | |
|---|---|---|---|---|
| | Chi-square | P-value | Chi-square | P-value |
| Sex | 10.07 | 0.002 | 15.05 | <0.001 |
| Ever had a terminated pregnancy | 0.24 | 0.625 | – | – |
| Twins | 43.62 | <0.001 | 21.97 | <0.001 |
| Gestation period | 48.20 | <0.001 | 13.69 | <0.001 |
| Tetanus vaccination before birth | 3.17 | 0.075 | 0.06 | 0.806 |
| Prenatal care | 2.67 | 0.102 | 0.13 | 0.717 |
| Wanted pregnancy when became pregnant | 3.22 | 0.073 | 0.06 | 0.803 |
| ANC | 0.27 | 0.600 | 0.07 | 0.789 |
| CS | 1.81 | 0.179 | 0.14 | 0.708 |
| Child size | 6.21 | 0.013 | 3.42 | 0.064 |
| Malaria in pregnancy | 8.82 | 0.003 | 0.10 | 0.756 |
| Mother needs intestinal drug during pregnancy | 3.69 | 0.055 | 0.01 | 0.911 |
| Child needs medical attention after delivery | 9.77 | 0.002 | 0.61 | 0.434 |
| Geographical location | 12.13 | <0.001 | – | – |
| Maternal educational level | 46.79 | <0.001 | 14.40 | <0.001 |
| Wealth index | 22.53 | <0.001 | 6.43 | 0.011 |
| Mother smoked during pregnancy | 0.06 | 0.806 | – | – |

violated the proportional hazards assumption. As shown in the top-left panel of Fig 3, the hazard coefficient begins to reverse or change direction at approximately 7 months. We used this time point to divide the survival time into two intervals (0–7 and 7–35 months) using the *survSplit* function from the *survminer* package in R software. This stratification approach was similarly applied to other covariates violating the assumption in both univariable and multivariable Cox-PH models.

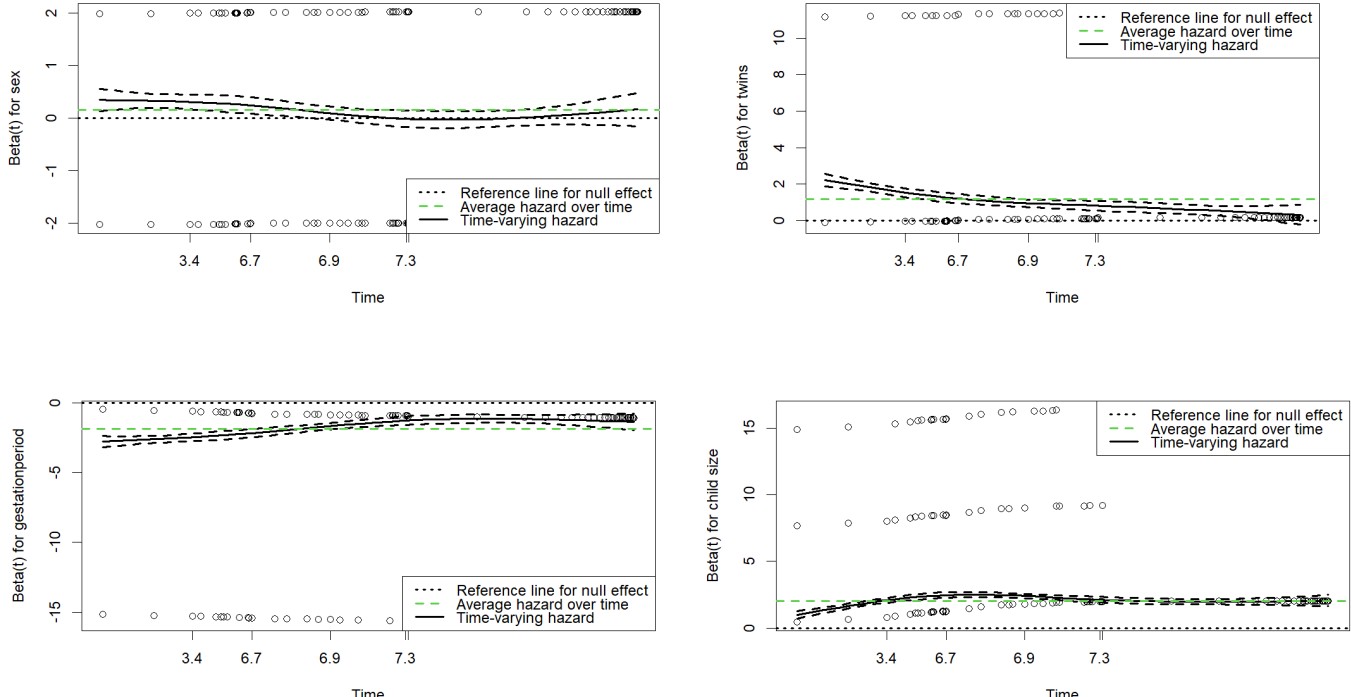

**Fig 3. The effects of covariates; sex (top-left panel), twins (top-right panel), gestation period (bottom-left panel) and child size (bottom-right panel) on death varies over time.**

Following the stratification, we refitted the extended Cox-PH model and reassessed the proportional hazards assumption. The results, presented in Table 3, confirm that all covariates in both the univariable and multivariable Cox-PH models now satisfy the assumption of proportional hazards over time. Tables 4 presents the unadjusted hazard ratio (unaHR) and adjusted hazard ratio (aHR) with their corresponding standard errors (s.e) and 95% confidence intervals (95% CI) from the extended or time-depended Cox-PH model for composite outcome and death and cardiac tamponade and, constriction respectively.

## Results from Bayesian survival model

In our dataset, under-five deaths constitute less than 10% of the total observations. This rarity can lead to biased model estimates, unstable or inflated hazard ratios, as highlighted in previous studies [42,41]. To address this issue, we employed Bayesian survival analysis [41,43], which has been shown to provide more accurate and stable estimates in low-event-rate scenarios. Parameter estimates from the Bayesian survival model were obtained via the Integrated Nested Laplace Approximation in R packaged INLA [23,24]. The results from the Bayesian survival model are presented in Table 5.

## Results from the multilevel survival model: frailty model

The adjusted hazard ratios from the multilevel survival model are presented in Table 6.

## Comparison of results from extended Cox-PH, Bayesian survival, and multilevel survival analysis models

We compared adjusted hazard ratios (aHRs) from the extended Cox-PH, Bayesian survival and the multilevel survival model (Table 7). While point estimates remained broadly consistent and statistical inferences from models remains

**Table 3. Checking Proportional Hazards Assumption.**

| | Unadjusted Cox-PH | | Adjusted Cox-PH | |
|---|---|---|---|---|
| Variable | Chi-square | P-value | Chi-square | P-value |
| Sex | 0.71 | 0.400 | 0.11 | 0.739 |
| Ever had a terminated pregnancy | 0.24 | 0.625 | – | – |
| Twins | 1.45 | 0.123 | 0.41 | 0.321 |
| Gestation period | 1.26 | 0.26 | 0.75 | 0.386 |
| Tetanus vaccination before birth | 3.17 | 0.075 | 0.06 | 0.806 |
| Parental care | 2.67 | 0.102 | 0.13 | 0.717 |
| Wanted pregnancy when became pregnant | 3.22 | 0.073 | 0.06 | 0.803 |
| ANC | 0.27 | 0.600 | 0.07 | 0.789 |
| CS | 1.81 | 0.179 | 0.14 | 0.708 |
| Child size | 2.44 | 0.290 | 3.42 | 0.064 |
| Malaria in pregnancy | 3.00 | 0.083 | 0.10 | 0.756 |
| Mother needed intestinal drug during pregnancy | 3.69 | 0.055 | 0.01 | 0.911 |
| Child needs medical attention after delivery | 1.78 | 0.180 | 0.61 | 0.434 |
| Geographical location | 0.03 | 0.86 | – | – |
| Maternal educational level | 3.25 | 0.071 | 1.15 | 0.285 |
| Wealth index | 3.31 | 0.069 | 0.84 | 0.3581 |
| Mother smoked during pregnancy | 0.06 | 0.806 | – | – |

unchanged, the Bayesian approach effectively removed inflated hazard ratios observed in the extended Cox-PH and multilevel survival models, providing more reliable estimates for rare under-five death events. Accordingly, all reported statistical results are based on the Bayesian survival model.

The results (Table 7) from the Bayesian survival model identified several significant predictors of time to death among children. Male children had a higher hazard of death compared to females (HR = 1.20, 95% CI: 1.11–1.30). Children from twin births were also at substantially increased risk relative to singletons (HR = 2.90, 95% CI: 2.51–3.34). In contrast, being born at term (≥9 months) was strongly protective, with a markedly reduced hazard of death compared to preterm births (HR = 0.16, 95% CI: 0.14–0.19). Caesarean delivery was associated with an increased hazard of death (HR = 1.60, 95% CI: 1.08–2.37).

Child size at birth was another important predictor. Both large-sized (HR = 3.82, 95% CI: 2.53–5.79) and average-sized (HR = 4.34, 95% CI: 2.92–6.44) children had significantly higher hazards of death compared to small-sized children. Conversely, children who required special attention after delivery had a reduced hazard of death (HR = 0.57, 95% CI: 0.38–0.85). Maternal socioeconomic factors also showed protective effects. Higher maternal education (HR = 0.59, 95% CI: 0.43–0.82) and secondary education (HR = 0.79, 95% CI: 0.71–0.88) were associated with reduced hazards of child death relative to mothers with no education. Similarly, belonging to middle-income (HR = 0.82, 95% CI: 0.72–0.92) and rich households (HR = 0.87, 95% CI: 0.78–0.98) conferred a survival advantage compared to poor households.

Overall, male sex, twin births, caesarean delivery, and larger birth size were associated with increased hazards of death, while term birth, maternal education, and higher household wealth offered significant protection.

## Discussion

In this study, we investigate the effects of potential predictors on early childhood mortality in Ghana using the extended Cox-PH model [22,33–35]. The extended Cox-PH model was used because certain predictors violated the assumption of constant covariate effects on hazard over time, as required by the standard Cox-PH model [36,37]. The data used in this study were obtained from the 2022 Ghana Demographic and Health Survey (GDHS 2022), publicly available at

**Table 4. Unadjusted and adjusted hazard ratio from the extended Cox-PH model for time-to-death.**

| Time to Death | unaHR | | | | aHR | | | |
|---|---|---|---|---|---|---|---|---|
| | HR | s.e | 95%CI | P-value | HR | s.e | 95%CI | P-value |
| **Sex** | | | | | | | | |
| Female (Ref) | – | – | – | – | – | – | – | – |
| Male | 1.28 | 0.05 | (1.14, 1.42) | <0.001 | 1.04 | 0.05 | (1.01, 1.30) | <0.001 |
| **Ever had a terminated pregnancy** | | | | | | | | |
| No (Ref) | – | – | – | – | – | – | – | – |
| Yes | 0.95 | 0.04 | (0.84,1.04) | 0.312 | – | – | – | – |
| **Twins** | | | | | | | | |
| Singletons (Ref) | – | – | – | – | – | – | – | – |
| Twins | 3.19 | 0.23 | (2.78,3.66) | <0.001 | 2.67 | 0.20 | (2.31,3.08) | <0.001 |
| **Gestation period** | | | | | | | | |
| <9 months (Ref) | – | – | – | – | – | – | – | – |
| ≥9 months | 0.10 | 0.09 | (0.08,0.12) | <0.001 | 0.20 | 0.69 | (0.17,0.23) | <0.001 |
| **Tetanus vaccination before birth** | | | | | | | | |
| No vaccination (Ref) | – | – | – | – | – | – | – | – |
| once | 40.28 | 6.22 | (29.76,54.51) | <0.001 | 2.53 | 0.66 | (1.54,4.18) | <0.001 |
| 2-7 times | 33.19 | 5.78 | (23.59,46.69) | <0.001 | 0.97 | 0.30 | (0.53,1.79) | 0.918 |
| **Parental care** | | | | | | | | |
| No (Ref) | – | – | – | – | – | – | – | – |
| Yes | 30.33 | 7.89 | (18.22,50.50) | <0.001 | 0.86 | 0.27 | (0.46,1.60) | 0.633 |
| **Wanted pregnancy when became pregnant** | | | | | | | | |
| Yes (Ref) | – | – | – | – | – | – | – | – |
| Later | 33.16 | 5.40 | (24.09,45.63) | <0.001 | 2.04 | 0.40 | (1.39,3.00) | <0.001 |
| No more | 38.69 | 14.68 | (18.39,81.39) | <0.001 | 0.53 | 0.23 | (0.23,1.23) | 0.140 |
| **ANC** | | | | | | | | |
| No (Ref) | – | – | – | – | – | – | – | – |
| Yes | 0.03 | 0.01 | (0.01,0.07) | <0.001 | 0.68 | 0.30 | (0.29,1.62) | 0.382 |
| **CS** | | | | | | | | |
| No (Ref) | – | – | – | – | – | – | – | – |
| Yes | 35.90 | 6.26 | (25.52,50.52) | <0.001 | 1.63 | 0.35 | (1.07,2.49) | 0.024 |
| **Child size** | | | | | | | | |
| Small (Ref) | – | – | – | – | – | – | – | – |
| Large | 17.15 | 0.25 | (10.43,28.20) | <0.001 | 9.97 | 2.23 | (6.43,15.46) | <0.001 |
| Average | 18.13 | 0.24 | (11.33,29.00) | <0.001 | 12.03 | 2.58 | (7.90,18.32) | <0.001 |
| **Malaria in pregnancy** | | | | | | | | |
| No (Ref) | – | – | – | – | – | – | – | – |
| Yes | 2.91 | 0.20 | (28.35,43.82) | <0.001 | 1.34 | 0.37 | (0.73,2.27) | 0.375 |
| **Mother needs intestinal drug during pregnancy** | | | | | | | | |
| No (Ref) | – | – | – | – | – | – | – | – |
| Yes | 35.70 | 5.27 | (26.72,47.68) | <0.001 | 1.34 | 0.32 | (0.84,2.14) | 0.221 |
| **Child needs attention after delivery** | | | | | | | | |
| No (Ref) | – | – | – | – | – | – | – | – |
| Yes | 5.57 | 0.41 | (2.49,12.44) | <0.001 | 1.02 | 0.23 | (0.65,1.60) | 0.924 |
| **Geographical location** | | | | | | | | |
| Rural (Ref) | – | – | – | – | – | – | – | – |
| Urban | 1.17 | 0.09 | (0.99,1.39) | 0.060 | – | – | – | – |

*(Continued)*

| Time to Death | unaHR | | | | aHR | | | |
|---|---|---|---|---|---|---|---|---|
| | HR | s.e | 95%CI | P-value | HR | s.e | 95%CI | P-value |
| **Maternal educational level** | | | | | | | | |
| No education (Ref) | – | – | – | – | – | – | – | – |
| Higher | 0.56 | 0.09 | (0.41,0.76) | <0.001 | 0.51 | 0.09 | (0.37,0.71) | <0.001 |
| Secondary | 0.78 | 0.04 | (0.71,0.86) | <0.001 | 0.76 | 0.04 | (0.68,0.84) | <0.001 |
| Primary | 1.01 | 0.06 | (0.91,1.13) | 0.861 | 1.01 | 0.06 | (0.90,1.13) | 0.845 |
| **Wealth index** | | | | | | | | |
| Poor (Ref) | – | – | – | – | – | – | – | – |
| Rich | 0.85 | 0.05 | (0.76,0.95) | 0.004 | 0.87 | 0.05 | (0.77,0.98) | 0.024 |
| Middle | 0.76 | 0.04 | (0.68,0.86) | <0.001 | 0.81 | 0.06 | (0.71,0.93) | 0.003 |
| **Mother smoked during pregnancy** | | | | | | | | |
| No (Ref) | – | – | – | – | – | – | – | – |
| Yes | 1.30 | 0.27 | (0.86,1.96) | 0.251 | – | – | – | – |

https://dhsprogram.com/methodology/survey/survey-display-598.cfm. Statistical analyses were carried out using R version 4.4.3 and STATA version 18.

Our findings indicate that male children have a higher risk of under-five mortality than females, which aligns with previous studies suggesting biological and genetic factors contribute to male vulnerability [43]. Male disadvantage in early life is repeatedly observed [24]. This increased risk may be linked to weaker immune responses and higher susceptibility to infections among male infants [23]. Twins exhibited a three-fold higher risk of mortality compared to singletons. This is consistent with studies demonstrating that twin pregnancies are often associated with preterm birth, low birth weight, and perinatal complications, which significantly increase mortality risks [24,25]. The increased burden on maternal resources and potential complications during delivery further contribute to these adverse outcomes [26]. Also, wins, who are more often preterm and low birthweight have substantially higher neonatal and under-five mortality across sub-Saharan Africa (SSA). Large multi-country DHS analyses and reviews consistently report excess twin mortality and the role of prematurity and growth restriction in this gap [44]. Infants born at full term (≥9 months) had an 84% lower risk of death than those born before 9 months. Premature birth is a well-documented risk factor for mortality, as preterm infants often suffer from respiratory distress syndrome, infections, and immature organ development, leading to higher morbidity and mortality rates [27]. The magnitude of protection in our study is consistent with estimates showing that complications of prematurity account for roughly half of neonatal deaths worldwide, especially where quality of care is variable [45,46]. These findings highlight the importance of improving antenatal care to prevent preterm births and associated complications.

Caesarean sections were linked to a 60% increase in under-five mortality. While caesarean sections are often life-saving, they may also expose infants to complications such as neonatal respiratory distress and increased infection risks [44]. The association between caesarean delivery and higher mortality revealed mixed findings from SAA; in many settings, CS are performed for life-threatening indications and can be markers of severity ("confounding by indication"); in addition, delays or quality gaps around intrapartum and immediate postnatal care may worsen outcomes for already-vulnerable newborns. Recent regional analyses [47,48] distinguish emergency versus elective procedures and link early neonatal deaths more strongly to emergency CS and weak health-system readiness, rather than the procedure itself. This finding suggests that unnecessary caesarean sections should be minimized, and postnatal care should be enhanced for caesarean section deliveries. Also, such findings are likely to be influenced by confounding factors. We acknowledge that residual confounding is possible due to unmeasured factors such as obstetric complications, comorbidities, or quality

**Table 5. Unadjusted and adjusted hazard ratio from the Bayesian survival model for time-to-death.**

| Variable | Category (Ref) | unaHR (95% CI) | aHR (95% CI) |
|---|---|---|---|
| Sex | Female (Ref) | – | – |
| | Male | 1.18 (1.08, 1.28) | 1.20 (1.11, 1.30) |
| Ever had a terminated pregnancy | No (Ref) | – | – |
| | Yes | 0.94 (0.86, 1.03) | – |
| Twins | Singletons (Ref) | – | – |
| | Twins | 3.31 (2.78, 3.66) | 2.90 (2.51, 3.34) |
| Gestation period | <9 months (Ref) | – | – |
| | ≥9 months | 0.15 (0.13, 0.19) | 0.16 (0.14, 0.19) |
| Tetanus vaccination before birth | No vaccination (Ref) | – | – |
| | Once | 4.23 (3.09, 5.80) | 1.05 (0.62, 1.79) |
| | 2–7 times | 2.91 (2.05, 4.14) | 0.71 (0.40, 1.27) |
| Parental care | No (Ref) | – | – |
| | Yes | 3.60 (2.15, 6.03) | 1.10 (0.62, 1.93) |
| Wanted pregnancy when became pregnant | Yes (Ref) | – | – |
| | Later | 4.82 (3.47, 6.70) | 1.44 (1.00, 2.09) |
| | No more | 3.04 (1.44, 6.43) | 0.55 (0.25, 1.21) |
| ANC | No (Ref) | – | – |
| | Yes | 0.12 (0.05, 0.27) | 0.48 (0.20, 1.12) |
| CS | No (Ref) | – | – |
| | Yes | 5.88 (4.14, 8.34) | 1.60 (1.08, 2.37) |
| Child size | Small (Ref) | – | – |
| | Large | 4.47 (3.38, 5.91) | 3.82 (2.53, 5.79) |
| | Average | 5.49 (4.17, 7.22) | 4.34 (2.92, 6.44) |
| Malaria in pregnancy | No (Ref) | – | – |
| | Yes | 3.92 (3.11, 4.96) | 1.50 (0.87, 2.59) |
| Mother needs intestinal drug during pregnancy | No (Ref) | – | – |
| | Yes | 3.29 (2.43, 4.44) | 0.99 (0.65, 1.50) |
| Child needs attention after delivery | No (Ref) | – | – |
| | Yes | 2.88 (2.17, 3.83) | 0.57 (0.38, 0.89) |
| Geographical location | Rural (Ref) | – | – |
| | Urban | 0.99 (0.91, 1.08) | – |
| Maternal educational level | No education (Ref) | – | – |
| | Higher | 0.60 (0.44, 0.82) | 0.59 (0.43, 0.82) |
| | Secondary | 0.80 (0.72, 0.88) | 0.79 (0.71, 0.88) |
| | Primary | 1.01 (0.91, 1.13) | 1.05 (0.94, 1.17) |
| Wealth index | Poor (Ref) | – | – |
| | Rich | 0.84 (0.75, 0.94) | 0.87 (0.78, 0.98) |
| | Middle | 0.77 (0.69, 0.86) | 0.82 (0.72, 0.92) |
| Mother smoked during pregnancy | No (Ref) | – | – |
| | Yes | 1.29 (0.86, 1.95) | – |

**Table 6. Adjusted hazard ratios from the multilevel survival model.**

| Variable | Category (Ref) | aHR (95% CI) |
|---|---|---|
| Sex | Female (Ref) | – |
| | Male | 1.20 (1.10, 1.30) |
| Twins | Singleton (Ref) | – |
| | Twin | 2.79 (2.41, 3.23) |
| Gestation period | Normal (Ref) | – |
| | Short | 0.18 (0.15, 0.21) |
| Tetanus vaccination before birth | None (Ref) | – |
| | 1 dose | 2.71 (1.63, 4.50) |
| | ≥2 doses | 1.00 (0.54, 1.86) |
| Prenatal care | No (Ref) | – |
| | Yes | 1.01 (0.54, 1.90) |
| Wanted pregnancy | No (Ref) | – |
| | Yes | 1.83 (1.23, 2.74) |
| | Later | 0.46 (0.18, 1.17) |
| ANC | No (Ref) | – |
| | Yes | 0.66 (0.27, 1.66) |
| CS | No (Ref) | – |
| | Yes | 1.94 (1.25, 3.00) |
| Child size at birth | Average (Ref) | – |
| | Small | 11.17 (7.15, 17.44) |
| | Large | 13.14 (8.59, 20.11) |
| Malaria in pregnancy | No (Ref) | – |
| | Yes | 1.47 (0.82, 2.63) |
| Intestine drug in pregnancy | No (Ref) | – |
| | Yes | 1.39 (0.86, 2.25) |
| Child needs attention | No (Ref) | – |
| | Yes | 0.81 (0.51, 1.28) |
| Maternal education | None (Ref) | – |
| | Primary | 0.54 (0.38, 0.75) |
| | Secondary | 0.82 (0.73, 0.93) |
| | Higher | 1.06 (0.94, 1.20) |
| Wealth index | Poor (Ref) | – |
| | Middle | 0.92 (0.81, 1.03) |
| | Rich | 0.91 (0.79, 1.04) |

of perinatal care. The observed association may reflect underlying risk profiles leading to emergency caesarean section rather than an adverse effect of caesarean delivery.

Birth weight was a significant predictor of mortality, with both average and large-sized infants showing a substantially increased risk. This contrasts with previous findings that primarily associate low birth weight with increased mortality [45,46,49]. However, the impact of child size on mortality risk gradually diminished over time, though it remained highly significant during the post-neonatal and early childhood periods [45]. The observed elevated risk among average- and large-sized infants may be linked to delivery complications, maternal diabetes, or unrecognized congenital conditions. In particular, infants in these size categories may include those affected by birth asphyxia, shoulder dystocia, or maternal conditions such as gestational diabetes and obstructed labour, all of which are known to increase neonatal morbidity and

**Table 7. Comparison of results from the extended Cox-PH, Bayesian survival and multilevel survival models for time-to-death.**

| Variable | Category (Ref) | Extended Cox-PH model: HR (95% CI) | Bayesian survival model: HR (95% CI) | Multilevel survival model: aHR (95% CI) |
|---|---|---|---|---|
| Sex | Female (Ref) | – | – | – |
| | Male | 1.04 (1.01, 1.30)** | 1.20 (1.11, 1.30)** | 1.20 (1.10, 1.30)** |
| Twins | Singleton (Ref) | – | – | – |
| | Twin | 2.67 (2.31, 3.08)** | 2.90 (2.51, 3.34)** | 2.79 (2.41, 3.23)** |
| Gestation period | <9 months (Ref) | – | – | – |
| | ≥9 months | 0.20 (0.17, 0.23)** | 0.16 (0.14, 0.19)** | 0.18 (0.15, 0.21)** |
| Tetanus vaccination before birth | None (Ref) | – | – | – |
| | Once/ 1 dose | 2.53 (1.54, 4.18)** | 1.05 (0.62, 1.79) | 2.71 (1.63, 4.50) |
| | 2–7 times/ ≥ 2 doses | 0.97 (0.53, 1.79) | 0.71 (0.40, 1.27) | 1.00 (0.54, 1.86) |
| Prenatal care | No (Ref) | – | – | – |
| | Yes | 0.86 (0.46, 1.60) | 1.10 (0.62, 1.93) | 1.01 (0.54, 1.90) |
| Wanted pregnancy | No (Ref) | – | – | – |
| | Later | 2.04 (1.39, 3.00)** | 1.44 (1.00, 2.09)** | 0.46 (0.18, 1.17) |
| | No more | 0.53 (0.23, 1.23) | 0.55 (0.25, 1.21) | 1.83 (1.23, 2.74) |
| ANC | No (Ref) | – | – | – |
| | Yes | 0.68 (0.29, 1.62) | 0.48 (0.20, 1.12) | 0.66 (0.27, 1.66) |
| CS | No (Ref) | – | – | – |
| | Yes | 1.63 (1.07, 2.49)** | 1.60 (1.08, 2.37)** | 1.94 (1.25, 3.00)** |
| Child size at birth | Small (Ref) | – | – | – |
| | Large | 9.97 (6.43, 15.46)** | 3.82 (2.53, 5.79)** | 13.14 (8.59, 20.11)** |
| | Average | 12.03 (7.90, 18.32)** | 4.34 (2.92, 6.44)** | 11.17 (7.15, 17.44)** |
| Malaria in pregnancy | No (Ref) | – | – | – |
| | Yes | 1.34 (0.73, 2.27) | 1.50 (0.87, 2.59) | 1.47 (0.82, 2.63) |
| Intestinal drug in pregnancy | No (Ref) | – | – | – |
| | Yes | 1.34 (0.84, 2.14) | 0.99 (0.65, 1.50) | 1.39 (0.86, 2.25) |
| Child needs attention | No (Ref) | – | – | – |
| | Yes | 1.02 (0.65, 1.60) | 0.57 (0.38, 0.85)** | 0.81 (0.51, 1.28) |
| Maternal education | No education (Ref) | – | – | – |
| | Higher | 0.51 (0.37, 0.71)** | 0.59 (0.43, 0.82)** | 1.06 (0.94, 1.20) |
| | Secondary | 0.76 (0.68, 0.84)** | 0.79 (0.71, 0.88)** | 0.82 (0.73, 0.93)** |
| | Primary | 1.01 (0.90, 1.13) | 1.05 (0.94, 1.17) | 0.54 (0.38, 0.75)** |
| Wealth index | Poor (Ref) | – | – | – |
| | Rich | 0.87 (0.77, 0.98)** | 0.87 (0.78, 0.98)** | 0.91 (0.79, 1.04)** |
| | Middle | 0.81 (0.71, 0.93)** | 0.82 (0.72, 0.92)** | 0.92 (0.81, 1.03)** |

** Statistical significance at 5%.

mortality [50–52]. Moreover, the size-at-birth measure used in this study is based on maternal recall rather than clinically measured birthweight, which may introduce misclassification bias [53]. In our dataset, birth size was self-reported by mothers rather than measured at delivery, which raises the possibility of recall bias and misclassification, as has been documented in similar Demographic and Health Survey (DHS) [54,55]. We also note that the categorical birth size variable may mask non-linear relationships between true birth weight and mortality risk. Measured continuous birth weight was not available in this dataset, hindering further modelling. These findings underscore the need for future research incorporating measured birthweight and detailed perinatal complication data to better understand this unexpected association [54,55].

An interesting finding from this study was that children who required special attention after delivery had a reduced hazard of death compared to those who did not. This result appears counterintuitive, as one might expect that newborns needing extra care would be at higher risk of mortality due to underlying complications. However, it is possible that these children benefited from closer monitoring and timely medical interventions immediately after birth, which improved their survival chances. Similar observations have been reported in some facility-based studies, where increased neonatal surveillance and prompt management of complications were associated with reduced early mortality [54,55]. In contrast, other studies have consistently shown that newborns requiring resuscitation or intensive care are at greater risk of death, reflecting the severity of underlying health conditions [47,56] (Mosley & Chen, 1984; Liu et al., 2016). Our finding may therefore reflect a context-specific effect, where the availability and quality of postnatal care played a decisive role. In settings with improved neonatal services, children identified as needing special attention may receive more comprehensive interventions, resulting in better outcomes than their apparently healthy counterparts who do not receive the same level of monitoring. This underscores the importance of strengthening postnatal care systems, particularly in resource-limited contexts, to ensure that vulnerable newborns receive timely and adequate attention.

Children from middle- and high-income families had a significantly lower risk of mortality, reinforcing the well-established relationship between socioeconomic status and child survival [48,57–59]. Wealthier families generally have better access to healthcare, nutrition, and safe living environments, which improve child survival rates. Wealth-related inequalities in under-five and neonatal mortality remain high in SAA, reflecting differences in access and quality of care across the continent. Estimates in this study align with the literature and underscore the value of multisectoral strategies that raise educational attainment and reduce financial barriers to timely, quality maternal-newborns care [63].

Furthermore, maternal education played a crucial role in reducing under-five mortality. Children of mothers with higher and secondary education had a 41% and 21% lower risk of death, respectively, compared to those whose mothers had no education. Educated mothers are more likely to access healthcare, practice proper child nutrition, and adopt hygienic practices, all of which contribute to improved child survival [57,54,55,60,61]. Large cross-country analyses attribute substantial global declines in child mortality to gains in women's schooling, operating through care-seeking, preventive practices, and household decision-making [61].

## Conclusion

This study identified key demographic, obstetric, and socioeconomic determinants of child survival. Male sex, twin births, caesarean delivery, and larger birth size were associated with increased hazards of death, while term gestation, maternal education, and higher household wealth offered significant protection. Interestingly, children identified as requiring special attention after delivery had improved survival, suggesting that early recognition and targeted postnatal care can mitigate risks. These findings highlight the need to strengthen maternal and newborn health systems, particularly intrapartum and immediate postnatal care, to address the vulnerabilities of twins, males, and caesarean-delivered infants. At the same time, investments in maternal education and poverty reduction remain critical for sustaining child survival gains. Strengthening health-system responsiveness and tackling structural inequalities will be essential to accelerate progress toward reducing under-five mortality

## Recommendation

Collectively, our findings recommend priorities to; prevent preterm birth where possible and ensure high-quality intrapartum and immediate new-born care, especially for twins and males; strengthen readiness and timeliness around emergency caesarean-sections; formalize postnatal special attention pathways so identification reliably triggers evidence-based care; and tackle structural drivers such as education and poverty that shape survival chances from birth. Where feasible, validating the birth-size result with measured birthweight and exploring indication-stratified CS effects would sharpen interpretation for policy. These targeted interventions are critical to sustaining progress toward reducing child mortality and achieving the Sustainable Development Goals.

## Limitation

This study has several limitations related to the use of DHS data. First, the cross-sectional design limits the ability to establish causal relationships between risk factors and child mortality. Second, many variables, including size at birth and pregnancy intention, are based on maternal recall and are subject to recall and reporting bias. Third, the DHS dataset lacks detailed clinical information, such as birthweight measurements, gestational age validation, and differentiation between elective and emergency caesarean sections, which restricts deeper interpretation of certain findings. Fourth, residual confounding may remain due to unmeasured factors such as quality of health services, maternal comorbidities, and intrapartum complications. Finally, mortality outcomes are reported retrospectively, which may lead to misclassification or underreporting of deaths, particularly in rural or hard-to-reach areas.

## Author contributions

**Conceptualization:** Emmanuel Boanyo.

**Data curation:** Abdul-Karim Iddrisu, Emmanuel Boanyo.

**Formal analysis:** Abdul-Karim Iddrisu, Emmanuel Boanyo.

**Methodology:** Abdul-Karim Iddrisu, Emmanuel Boanyo.

**Resources:** Abdul-Karim Iddrisu, Emmanuel Boanyo.

**Software:** Abdul-Karim Iddrisu, Emmanuel Boanyo.

**Supervision:** Abdul-Karim Iddrisu.

**Visualization:** Abdul-Karim Iddrisu, Emmanuel Boanyo.

**Writing – original draft:** Abdul-Karim Iddrisu, Emmanuel Boanyo.

**Writing – review & editing:** Abdul-Karim Iddrisu, Emmanuel Boanyo.

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
