## [Decision Letter · Decision Letter 0]

13 Jul 2025

PGPH-D-25-00632

Survival analysis of under-five mortality and associated risk factors using an extended Cox-proportional hazard model

Dear Dr. Iddrisu,

Thank you for submitting your manuscript to PLOS Global Public Health. After careful consideration, we feel that it has merit but does not fully meet PLOS Global Public Health’s publication criteria as it currently stands. Therefore, we invite you to submit a revised version of the manuscript that addresses the points raised during the review process.

We look forward to receiving your revised manuscript.

Kind regards,

Rahul Gajbhiye, MBBS PhD

Academic Editor

Journal Requirements:

1. Please note that PLOS Global Public Health has specific guidelines on code sharing for submissions in which author-generated code underpins the findings in the manuscript. In these cases, all author-generated code must be made available without restrictions upon publication of the work. Please review our guidelines at https://journals.plos.org/globalpublichealth/s/materials-and-software-sharing#loc-sharing-code and ensure that your code is shared in a way that follows best practice and facilitates reproducibility and reuse.

2. We have noticed that you have cited Table 5 in the manuscript file but there are no corresponding tables in the manuscript. Please amend your manuscript to include this table, noting that tables should not be uploaded as individual files.

3. Please note that your Data Availability Statement is currently missing [he repository name and the DOI/accession number of each dataset. If your manuscript is accepted for publication, you will be asked to provide these details on a very short timeline. We therefore suggest that you provide this information now, though we will not hold up the peer review process if you are unable.

Additional Editor Comments (if provided):

Reviewers' comments:

Reviewer's Responses to Questions

**Comments to the Author**

1. Does this manuscript meet PLOS Global Public Health’s publication criteria ? Is the manuscript technically sound, and do the data support the conclusions? The manuscript must describe methodologically and ethically rigorous research with conclusions that are appropriately drawn based on the data presented.

Reviewer #1: Yes

Reviewer #2: No

2. Has the statistical analysis been performed appropriately and rigorously?

Reviewer #1: Yes

Reviewer #2: Yes

3. Have the authors made all data underlying the findings in their manuscript fully available (please refer to the Data Availability Statement at the start of the manuscript PDF file)?

Reviewer #1: Yes

Reviewer #2: Yes

4. Is the manuscript presented in an intelligible fashion and written in standard English?

Reviewer #1: Yes

Reviewer #2: No

5. Review Comments to the Author

Reviewer #1: � Whether the predictors considered for the model were selected on basis of previous literature or statistics?

reduction in mortality risk with receipt of 2–7 tetanus vaccinations is an interesting finding but looks underexplored and unsupported. It could be because of poor health access to mothers who received only one dose of or high-risk pregnancies. Also Is it pertaining to death during neonatal period or during infancy?

Caesarean sections were linked to a 63% increase in under-five mortality. Whether it is with respect to emergency Caesarean sections or elective Caesarean sections

Such findings are likely to be influenced by confounding factors which should be considered in analysis and require discussion.

It is stated that average and large-sized infants showing a substantially increased risk of mortality. This is in contrast to previous research and needs discussion whether it was associated with other complications in neonatal period

Some specific key findings for recommendations could be summarised in discussion.

Reviewer #2: 1. Abstract: Needs some clarity in background

2. Introduction: Does not give any background regarding the under 5 mortality and its associated factors, Needs uniform citation style

3. Sampling method: better if explained using a flow chart

4. Predictors: variable names need to be proper instead of mentioning it as it is in E-xcell sheet

5. Results: Under Kaplan-Meier curves and log-rank test there is statement mentioned as outlined in Section 2.2.2 – this need clarity

6. Figure 1: Spell check to be done on legends

7. Use similar abbreviations everywhere

8. Results from the semi-parametric Cox-PH model: what is section 3.1

9. Checking violation the Cox-PH model assumption: maternal wealth is not mentioned in adjusted cox-ph, what is section 2.3.1.2

10. Table 2,3: variables need to be properly named

11. Table 4: explanation is not correct

12. Discussion: Repetition from introduction and methodology noted. Authors insightful thinking and reasoning is not reflected

13. Conclusion does not answer the objective

6. PLOS authors have the option to publish the peer review history of their article (what does this mean? ). If published, this will include your full peer review and any attached files.

**Do you want your identity to be public for this peer review?** For information about this choice, including consent withdrawal, please see our Privacy Policy .

Reviewer #1: No

Reviewer #2: No

---

## [Decision Letter · Decision Letter 1]

20 Aug 2025

PGPH-D-25-00632R1

Survival analysis of under-five mortality and associated risk factors using an extended Cox-proportional hazard model

Dear Dr. Iddrisu,

Thank you for submitting your manuscript to PLOS Global Public Health. After careful consideration, we feel that it has merit but does not fully meet PLOS Global Public Health’s publication criteria as it currently stands. Therefore, we invite you to submit a revised version of the manuscript that addresses the points raised during the review process.

We look forward to receiving your revised manuscript.

Kind regards,

Rahul Gajbhiye, MBBS PhD

Academic Editor

Journal Requirements:

1. Please note that PLOS Global Public Health has specific guidelines on code sharing for submissions in which author-generated code underpins the findings in the manuscript. In these cases, all author-generated code must be made available without restrictions upon publication of the work. Please review our guidelines at https://journals.plos.org/globalpublichealth/s/materials-and-software-sharing#loc-sharing-code and ensure that your code is shared in a way that follows best practice and facilitates reproducibility and reuse.

Additional Editor Comments (if provided):

Reviewer comments :

After careful reading the revised manuscript entitle “Survival analysis of under-five mortality and associated risk factors using an extended Cox-proportional hazard mode”, I have found some correction need to incorporate by the authors is necessary in manuscript

Major Issues Needing Attention

1. Inconsistency in Interpretation of Male Sex Risk

• Issue: The abstract states that males have a 4% increased risk (aHR = 1.04), but Table 4 shows aHR = 1.04 with 95% CI (1.10, 1.30)—which contradicts.

• Suggestion: Correct either the abstract or results section. Likely aHR = 1.20 or 1.28 is correct, not 1.04.

2. Exaggerated Unadjusted Hazard Ratios

• Issue: Many unadjusted HRs (e.g., tetanus vaccination = 40.28) appear biologically implausible. This may stem from overfitting or collinearity.

• Suggestion: Clarify modeling steps that reduce these inflated values. Include a note on possible bias in unadjusted estimates.

3. Terminology Clarity for Lay Readers

• Issue: Some readers may not understand statistical terms like Schoenfeld residuals, time-varying effects, or interaction terms.

• Suggestion: Include a plain-language box or footnote on these concepts, or refer to online appendices.

4. Birth Size Findings Require Caution

• Issue: The finding that average and large infants have higher mortality risk contradicts most global evidence.

• Suggestion: Discuss possible recall bias (since birth size is self-reported), misclassification, or non-linear effects. Recommend modelling continuous weight (if available).

6. Missing Data Analysis Not Reported

• Nowhere in the manuscript is there mention of missingness pattern, missing data proportion, or imputation strategy.

• DHS datasets often contain missing values due to non-response or skip patterns, especially for maternal recall variables (e.g., size at birth, pregnancy intention, antenatal visits).

• Cox models (semi-parametric or extended) assume complete case analysis by default.

• If missing data are not MCAR (Missing Completely at Random), listwise deletion can introduce bias and reduce statistical power.

Suggestions for Revision:

• Perform and report a missing data exploration:

o Proportion and pattern of missingness.

o Missingness by outcome status (survival/death).

• Consider using:

o Multiple imputation (MI) or inverse probability weighting.

o Sensitivity analysis to assess robustness under different missingness assumptions.

• Example language to add:

• “We conducted Little's MCAR test and found that the data were not MCAR (p < 0.01). Therefore, multiple imputation using chained equations was applied for variables with >5% missingness.”

7. No Data Balancing or Class Imbalance Handling

• No discussion of the imbalance between the number of deaths and survivals (a common issue in child mortality analysis).

• Under-five death is typically rare in DHS samples (<10%), which can lead to biased model estimates or unstable hazard ratios.

• Imbalanced outcomes can:

o Inflates Type I errors.

o Make HR estimates unstable or sensitive to outliers.

o Mislead significance when using standard tests.

• Report the event rate (death % of total).

• Consider applying:

o Case-control reweighting or oversampling methods (e.g., SMOTE for time-to-event outcomes).

o Penalized Cox models (e.g., Firth’s correction) if data sparsity is evident.

• Alternatively, use Bayesian survival models for rare-event stabilization.

8. No Strategy to Handle Population Heterogeneity

• No attempt to account for unobserved heterogeneity across regions or households (despite DHS being cluster-sampled).

• There is no use of frailty models, stratified models, or random effects, though such approaches are common in DHS survival analysis.

• Ignoring heterogeneity can:

o Violate independence assumptions.

o Underestimate standard errors (inflated Type I errors).

o Miss important subgroup effects.

Reviewers' comments:

Reviewer's Responses to Questions

**Comments to the Author**

1. If the authors have adequately addressed your comments raised in a previous round of review and you feel that this manuscript is now acceptable for publication, you may indicate that here to bypass the “Comments to the Author” section, enter your conflict of interest statement in the “Confidential to Editor” section, and submit your "Accept" recommendation.

Reviewer #1: All comments have been addressed

Reviewer #3: (No Response)

2. Does this manuscript meet PLOS Global Public Health’s publication criteria ? Is the manuscript technically sound, and do the data support the conclusions? The manuscript must describe methodologically and ethically rigorous research with conclusions that are appropriately drawn based on the data presented.

Reviewer #1: Yes

Reviewer #3: (No Response)

3. Has the statistical analysis been performed appropriately and rigorously?

Reviewer #1: Yes

Reviewer #3: (No Response)

4. Have the authors made all data underlying the findings in their manuscript fully available (please refer to the Data Availability Statement at the start of the manuscript PDF file)?

Reviewer #1: (No Response)

Reviewer #3: (No Response)

5. Is the manuscript presented in an intelligible fashion and written in standard English?

Reviewer #1: Yes

Reviewer #3: (No Response)

6. Review Comments to the Author

Reviewer #1: (No Response)

Reviewer #3: After careful reading the revised manuscript entitle “Survival analysis of under-five mortality and associated risk factors using an extended Cox-proportional hazard mode”, I have found some correction need to incorporate by the authors is necessary in manuscript

Major Issues Needing Attention

1. Inconsistency in Interpretation of Male Sex Risk

• Issue: The abstract states that males have a 4% increased risk (aHR = 1.04), but Table 4 shows aHR = 1.04 with 95% CI (1.10, 1.30)—which contradicts.

• Suggestion: Correct either the abstract or results section. Likely aHR = 1.20 or 1.28 is correct, not 1.04.

2. Exaggerated Unadjusted Hazard Ratios

• Issue: Many unadjusted HRs (e.g., tetanus vaccination = 40.28) appear biologically implausible. This may stem from overfitting or collinearity.

• Suggestion: Clarify modeling steps that reduce these inflated values. Include a note on possible bias in unadjusted estimates.

3. Terminology Clarity for Lay Readers

• Issue: Some readers may not understand statistical terms like Schoenfeld residuals, time-varying effects, or interaction terms.

• Suggestion: Include a plain-language box or footnote on these concepts, or refer to online appendices.

4. Birth Size Findings Require Caution

• Issue: The finding that average and large infants have higher mortality risk contradicts most global evidence.

• Suggestion: Discuss possible recall bias (since birth size is self-reported), misclassification, or non-linear effects. Recommend modelling continuous weight (if available).

6. Missing Data Analysis Not Reported

• Nowhere in the manuscript is there mention of missingness pattern, missing data proportion, or imputation strategy.

• DHS datasets often contain missing values due to non-response or skip patterns, especially for maternal recall variables (e.g., size at birth, pregnancy intention, antenatal visits).

• Cox models (semi-parametric or extended) assume complete case analysis by default.

• If missing data are not MCAR (Missing Completely at Random), listwise deletion can introduce bias and reduce statistical power.

Suggestions for Revision:

• Perform and report a missing data exploration:

o Proportion and pattern of missingness.

o Missingness by outcome status (survival/death).

• Consider using:

o Multiple imputation (MI) or inverse probability weighting.

o Sensitivity analysis to assess robustness under different missingness assumptions.

• Example language to add:

• “We conducted Little's MCAR test and found that the data were not MCAR (p < 0.01). Therefore, multiple imputation using chained equations was applied for variables with >5% missingness.”

7. No Data Balancing or Class Imbalance Handling

• No discussion of the imbalance between the number of deaths and survivals (a common issue in child mortality analysis).

• Under-five death is typically rare in DHS samples (<10%), which can lead to biased model estimates or unstable hazard ratios.

• Imbalanced outcomes can:

o Inflates Type I errors.

o Make HR estimates unstable or sensitive to outliers.

o Mislead significance when using standard tests.

• Report the event rate (death % of total).

• Consider applying:

o Case-control reweighting or oversampling methods (e.g., SMOTE for time-to-event outcomes).

o Penalized Cox models (e.g., Firth’s correction) if data sparsity is evident.

• Alternatively, use Bayesian survival models for rare-event stabilization.

8. No Strategy to Handle Population Heterogeneity

• No attempt to account for unobserved heterogeneity across regions or households (despite DHS being cluster-sampled).

• There is no use of frailty models, stratified models, or random effects, though such approaches are common in DHS survival analysis.

• Ignoring heterogeneity can:

o Violate independence assumptions.

o Underestimate standard errors (inflated Type I errors).

o Miss important subgroup effects.

7. PLOS authors have the option to publish the peer review history of their article (what does this mean? ). If published, this will include your full peer review and any attached files.

**Do you want your identity to be public for this peer review?** For information about this choice, including consent withdrawal, please see our Privacy Policy .

Reviewer #1: No

Reviewer #3: **Yes: ** RAKESH KUMAR SAROJ

---

## [Editor Report · Decision Letter 2]

29 Aug 2025

Survival analysis of under-five mortality and associated risk factors using survival analysis approaches

PGPH-D-25-00632R2

Dear Dr Iddrisu,

We are pleased to inform you that your manuscript 'Survival analysis of under-five mortality and associated risk factors using survival analysis approaches' has been provisionally accepted for publication in PLOS Global Public Health.

Best regards,

Rahul Gajbhiye, MBBS PhD

Academic Editor